



# Millennial variability of terrigenous transport to the central-southern Peruvian margin during the last deglaciation (18-13 kyr BP)

Marco Yseki[1], Bruno Turcq[1], Sandrine Caquineau[1], Renato Salvatteci[2], José Solis[3], C. Gregory Skilbeck[4], Dimitri Gutiérrez[3,5]

[1]LOCEAN-IPSL, Laboratoire d'Océanographie et du Climat: Expérimentation et Approches Numériques, Sorbonne Université, CNRS, IRD, MNHN, Paris, France.
[2]Center for Ocean and Society, Kiel University, Kiel, 24105, Germany.
[3]Laboratorio de Ciencias del Mar, Facultad de Ciencias y Filosofía, Universidad Peruana Cayetano Heredia, Lima, Peru.
[4]Faculty of Science, University of Technology Sydney. PO Box 123 Broadway, Sydney NSW. 2007.
[5]Dirección General de Investigaciones Oceanográficas y de Cambio Climático, Instituto del Mar del Perú, Callao. Peru.

*Correspondence to*: Marco Yseki (marco.yseki@gmail.com)

**Abstract.** Reconstructing precipitation and wind from the geological record could help to understand the potential changes in precipitation and wind dynamics in response to climate change in Peru. The last deglaciation offers natural experimental
conditions to test precipitation and wind dynamics response to high latitude forcing. While considerable research has been done to reconstruct precipitation variability during the last deglaciation in the Atlantic sector of South America, the Pacific sector of South America has received little attention. This work aims to fill this gap by reconstructing types of terrigenous transport to the central-southern Peruvian margin (12°S and 14ºS) during the last deglaciation (18-13 kyr BP). For this purpose, we used grain-size distribution in sediments of marine core M77/2-005-3 (Callao, 12ºS) and G14 (Pisco, 14ºS). We analyzed
end-members (EM) to identify grain-size components and reconstruct potential sources and transport processes of terrigenous material across time. We identified four end-members for both Callao and Pisco sediments. In Callao, we propose that changes in EM4 (101 μm) and EM2 (58 μm) contribution mainly reflect hydrodynamic energy and diffuse sources, respectively, while EM3 (77 um) and EM1 (11 μm) variations reflect changes in aeolian and fluvial inputs, respectively. In Pisco, changes in the contribution of EM1 (10 μm) reflect changes in river inputs while EM2 (52 μm), EM3 (75 μm) and EM4 (94 μm) reflect an
aeolian origin linked to surface winds. At millennial-scale, our record shows an increase of the fluvial inputs during the last part of Heinrich Stadial 1 (~ 16-14.7 kyr BP) at both locations. This increase was linked to higher precipitation in Andes related to a reduction of the Atlantic Meridional Overturning Circulation and meltwater discharge in North Atlantic. In contrast, during Bølling-Allerød (~ 14.7-13 kyr BP), there was an aeolian input increase, associated with stronger winds and lower precipitation that indicate an expansion of the South Pacific Subtropical High. These conditions would correspond to a northern
displacement of the Intertropical Convergence Zone-South Subtropical High system associated with a stronger Walker circulation. Our results suggest that variations in river discharge and changes in surface wind intensity in the western margin of South America during the last deglaciation were sensitive to Atlantic Meridional Overturning Circulation variations and Walker circulation on millennial timescales. In the context of global warming, large-scale precipitation and fluvial discharge





increases in the Andes related to Atlantic Meridional Overturning Circulation decline and southward displacement of the
Intertropical Convergence Zone should be considered.

## 1. Introduction

The last deglaciation, a period of global warming from the end of the Last Glacial Maximum (LGM, ~ 19 kyr BP) to the early
Holocene (11.7 kyr BP), is an outstanding period in Earth's history that allows us a better understanding of the mechanisms
regulating regional climatic conditions under global warming (Clark et al., 2012; Shakun et al., 2012). During the last
deglaciation, variations in meltwater discharge in the North Atlantic and their consequent impact on the intensity of the Atlantic
Meridional Overturning Circulation (AMOC) generated abrupt climatic changes on a millennial-scale (McManus et al., 2004;
Mulitza et al., 2017; Ng et al., 2018). The latter resulted in changes in the meridional-oceanic temperature gradient and a
meridional shift of the mean annual position of the Intertropical Convergence Zone (ITCZ) (Cheng et al., 2012; Deplazes et
al., 2013; Mcgee et al., 2014).
Numerous studies based on continental and marine records have evaluated the effects of meltwater discharge and temperature
variations in the North Atlantic on precipitation in Tropical South America, TSA (e.g., Mollier-Vogel et al., 2013; Novello et
al., 2017; Mulitza et al., 2017; Stríkis et al., 2015, Bahr et al., 2018). Most studies suggest wetter conditions in this region
during cold events in North Hemisphere such as Heinrich Stadial 1 (HS1, ~ 18-14.7 kyr BP) and the Younger Dryas
(YD, ~ 12.9-11 kyr BP) linked to a southern displacement of the ITCZ (e.g., Mollier-Vogel et al., 2013; Mulitza et al., 2017;
Bahr et al., 2018) and an intensification of the South American Monsoon in its southern domain (Novello et al., 2017; Stríkis
et al., 2015) in response to the weakening of the AMOC and increased meltwater discharges into the North Atlantic.
Conversely, during the Bølling-Allerød (B-A, 14.7-12.9 kyr BP), a warm period in North Hemisphere, dry conditions
developed in TSA (e.g., Mollier-Vogel et al., 2013; Novello et al., 2017; Mulitza et al., 2017) associated with a strong AMOC,
a more northerly position of the ITCZ and a weakening of the South American Monsoon.
However, most records covering the last deglaciation come from Eastern South America (e.g., Cruz et al., 2005; Stríkis et al.,
2015; Montade et al., 2015; Zhang et al., 2015; Bahr et al., 2018; Novello et al., 2017, Mulitza et al., 2017; Stríkis et al., 2018),
while records from the western slope of the Andes (e.g., Baker et al., 2001a, 2001b) and the Peruvian margin are scarce (e.g.,
Rein et al., 2005; Mollier-Vogel et al., 2013). Previous attempts to reconstruct changes in precipitation in the western flank of
the Andes using marine sediment records show contrasting results. In northern Peru (4°S), Mollier-Vogel et al. (2013) based
on Titanium (Ti) to Calcium (Ca) ratios, suggested an increase in fluvial inputs during HS1 and YD and reduced precipitation
during the B-A. However off Callao (12°S), no difference in fluvial inputs, based on lithic content, between HS1 and BA was
reported (Rein et al., 2005). The difference between both records could be due to changes in sediment transport at the two sites
and/or to the interpretation of proxies used in these studies. In both studies, Ti/Ca at 4°S (Mollier-Vogel et al., 2013) and lithic
content at 12°S (Rein et al., 2005) are interpreted as indicators of fluvial inputs. The latter, is generally true in northern Peru,
4°S, where rainfall can reach 466 mm y$^{-1}$ (Lagos et al., 2008). However, other processes can be invoked in more arid regions



such as central-southern Peru where rainfall is scarce (less than 20 mm y$^{-1}$) (Lagos et al., 2008). Indeed, Briceño-Zuluaga et al. (2016) showed that, during the last millennium, part of the detrital fraction of marine sediments collected off Pisco was also of aeolian origin. According to Briceño-Zuluaga et al. (2016) aeolian inputs off Pisco can contribute up to almost 50% of the terrigenous fraction during some climatic periods (e.g., the Medieval Climatic Anomaly). These results are based on the grain

size distributions of terrigenous components in the sediment.

The grain-size distribution of Peruvian margin sediments is typically polymodal and for that reason provides information on sediment transport mechanisms and/or sediment sources (Briceño-Zuluaga et al., 2016). Aeolian particles diameters are relatively coarser than fluvial ones and if wind intensification occurs, the aeolian flow and the frequency of coarse particles (~ >36 μm) would increase. Thus, the relative abundance of fluvial particles (~ 6-14 μm) would reflect the precipitation and

continental runoff (e.g., Stuut et al., 2002; Stuut and Lamy 2004; Pichevin et al., 2005; Briceño-Zuluaga, et al., 2016; Beuscher et al., 2017). Mathematical methods are used to identify grain-size components of polymodal sediments. For instance, End Member Analysis (EMA) has been widely used to infer changes in fluvial and/or aeolian inputs (e.g., Stuut et al., 2002, 2004, 2007, 2014; Weltje y Prins, 2003; 2007; Pichevin et al., 2005; Holz et al., 2007; Just et al., 2012; Beuscher et al., 2017; Humphries et al., 2017; Jiang et al., 2017).

The aim of this work is to reconstruct at millennial-scale the transport (fluvial and aeolian) and sedimentation of the terrigenous inputs off central-southern Peru (Callao and Pisco) during the last deglaciation. For this purpose, grain size distributions on surface sediments and sediments cores (M77/2-005-3, Callao and G14, Pisco) were measured and EMA was used to deconvolved them into sub-populations. Using this methodology allows us to discriminate the different sources of terrigenous input, reconstructing the variations of fluvial and aeolian inputs dependent on changes in precipitation and winds intensity. On

other hand, surface sediments were collected for grain size analyses during normal conditions and during the 2017 Coastal El Niño (April 2017), the latter characterized by increased river discharges (Guzman et al., 2020) and variations of surface winds in Peru (Echevin et al., 2018). These observations allow us to understand the effect of changes in precipitation and winds on grain-size distribution during periods of different climatic conditions. In addition, we used as a proxy for fluvial vs aeolian inputs the Titanium/Zirconium (Ti/Zr) record from X-Ray Fluorescence (XRF) analysis of the 106KL core collected off Callao

and described by Rein et al. (2005). We postulate that changes in the AMOC intensity have modulated the variability of winds and precipitation in the Western TSA, as inferred by changes in the grain size distribution of marine sediment particles, at millennial time-scales. Our work provides new information on sedimentation, types of transport and sources of terrigenous inputs on the Peruvian margin during the last deglaciation, offering the possibility to better understand the mechanisms modulating these processes during past periods of global warming.

## 1.2 Regional setting


We focus on the central-southern part (12-14ºS) of the Peruvian margin. Callao and Pisco are located onshore of the Lima Basin (Suess et al., 1987). This basin exhibits high productivity and anoxic conditions favored by an intense Oxygen Minimum Zone; hence sediments are composed of fine grains, are rich in organic matter and contain abundant diatoms. The general





absence of bioturbation in some areas and during some time periods allows the preservation of laminations and therefore their
use as palaeoceanographic records (e.g., Rein et al., 2005; Sifeddine et al., 2008; Gutiérrez et al., 2006, 2009, 2011; Briceño-
Zuluaga et al., 2016; Salvatteci et al., 2014a, 2016, 2019). In Callao, muddy laminated areas are reported (Reinhardt et al.,
2002), but sedimentary records collected in the OMZ core, off Pisco, show more continuous laminations than the records
collected off Callao (e.g., Salvatteci et al., 2016, 2019).

The main transport of the detrital fraction of coarse silt and sand to the hemipelagic sediments in the Peruvian margin is by the
action of winds (Scheidegger and Krissek, 1982). In contrast to Callao, Pisco is characterized by the presence of large coastal
deserts, extreme aridity and dust storms known as Paracas winds. During these sporadic sand storms wind velocities can exceed
10-15 m/s (Briceño-Zuluaga et al., 2017). These storms are produced by a local intensification of alongshore surface wind and
by alongshore pressure gradients (Briceño-Zuluaga et al., 2017). Also, in Pisco, an intense coastal upwelling linked to strong
alongshore surface wind occurs (Dewitte et al., 2011; Gutiérrez et al., 2011; Rahn an Garreaud, 2013). The intensity of
alongshore surface winds presents a seasonal variability, with stronger winds during winter and weaker winds during summer
(Fig. 1). This seasonality is linked regionally to the displacements of the ITCZ-South Pacific Subtropical High (SPSH) system
and locally to continental-oceanic and alongshore pressure gradients (Strub et al., 1998; Gutiérrez et al., 2011; Chamorro et
al., 2018). In contrast to the coarser particles transported by winds, large quantities of quartz-rich silt and clays are transported
by rivers to the continental shelf (Scheidegger and Krissek, 1982). The central-southern Peruvian coast is characterized by
very low annual precipitations (Callao,14 mm y$^{-1}$ and Pisco, 2 mm y$^{-1}$) and scarce flows of coastal rivers (Lagos et al., 2008).
However, during summer there is an increase in river discharges associated with increased monsoon precipitation in the Andes
(Garreaud et al., 2009; Vuille et al., 2012). Occasional floods occur and higher sediment discharges are associated with intense
precipitation during extreme El Niño events (Bourrel et al., 2015; Morera et al., 2017; Rau et al., 2016; Guzman et al., 2020).
Aeolian clays and silts are transported offshore by trade winds (Saukel et al., 2011), while coarser particles (e.g., > 40 μm)
settle in the ocean on the continental shelf (Scheidegger and Krissek, 1982). Once in the water column, dispersion patterns of
clays (< 4 μm) and fine silts (8-11 μm) coincide with the surface and subsurface currents, while the coarser fraction presents
limited dispersion near the coast (Scheidegger and Krissek, 1982). Likewise, near-bottom processes and bottom topography
exert considerable control over the dispersal of hemipelagic sediments on the Peruvian margin (Scheidegger and Krissek,
1982).

## 2. Materials and methods

### 2.1 Surface sediments, marine core and age model

Surface sediments (0-0.5 cm) were collected in front of Callao and Pisco by the Instituto del Mar del Peru during the years
2015 (December), 2016 (August and December) and 2017 (February, April and August) along transects perpendicular to the
coast (Fig. S1). Details on the sampling sites are given in Table S1. Samples collected in April 2017 coincided with the



occurrence of a coastal El Niño event, and will be considered as representative of "Coastal El Niño" conditions hereafter. All the other samples will represent "normal" conditions.

The core M77/2-005-3, was retrieved from the Southeast Pacific continental slope (12º05 S, 77º40,07 W, 214 m water depth, 1336 cm long) during the M77-2 expedition in 2008 (Fig. 1). Because we focus on the last deglaciation period, we worked with the section from 0 to 700 cm core depth. A first depth-age model based on four 14C ages was built in Salvatteci et al.

(2019). In this study, we added twenty-two [14]C ages and developed a new age depth-model (Table S2). Radiocarbon measurements were performed on bulk sediment at the Laboratoire de Mesures du Carbone-14 (LMC14, Gif-sur Yvette, France). Ortlieb et al. (2011) reported a regional reservoir effect (ΔR) of 511 ± 278 years for Early Holocene (10.4-6.8 kyr) and in the absence of ΔR data for older period we use this value to calibrated the 14C measurements for the last deglaciation as in Salvatteci et al. (2016, 2019). In order to construct the age model, we used the maximum probability ages obtained from

the CALIB 8.1 software. The chronological model indicates that the examined section (0-700 cm) of core M77/2-005-3 (Fig. S2) recorded the LGM and the last deglaciation (22-13 kyr BP; 95-700 cm). Core M77/2-005-3 presents a hiatus at 94 cm and thus great part of the Holocene is missing.

Core G14 (14ºS, 76ºW, 390 m water depth) was retrieved during the Galathea-3 expedition in 2007 (Salvatteci et al., 2016). Radiocarbon dates of G14 were published in Salvatteci et al. (2016). For this study, a new depth-age model was developed

with an updated calibration using the CALIB 8.1 software. The upper part of sediment layers was not recovered in G14 which ranges from 13.4 to 24.6 kyr BP (Fig. S2).

The lithological description of M77/2-005-3 and G14 is available in Salvatteci et al. (2016, 2019). M77/2-005-3 and G14 cores show laminated and banded sediments with no evident signs of major discontinuities during the last deglaciation (Salvatteci et al., 2016, 2019). G14 core presents more continuous laminated packaged in comparison to M722-005-3 core. (Salvatteci et

al., 2016, 2019).

In order to compare our new data from cores M77/2-005-3 and G14 with previously published records in the area we modified the age model of core 106 KL (Rein et al., 2005). Core 106 KL (12°030S, 77°39.80W, 184 m water depth) was retrieved during cruise SONNE 147 (Rein et al., 2004, 2005). Chronology model and lithology were fully described in Rein et al. (2004, 2005). A new depth-age model was developed with an updated calibration using the CALIB 8.1 software (Fig. S2) and we

only used the sections covering the last deglaciation.





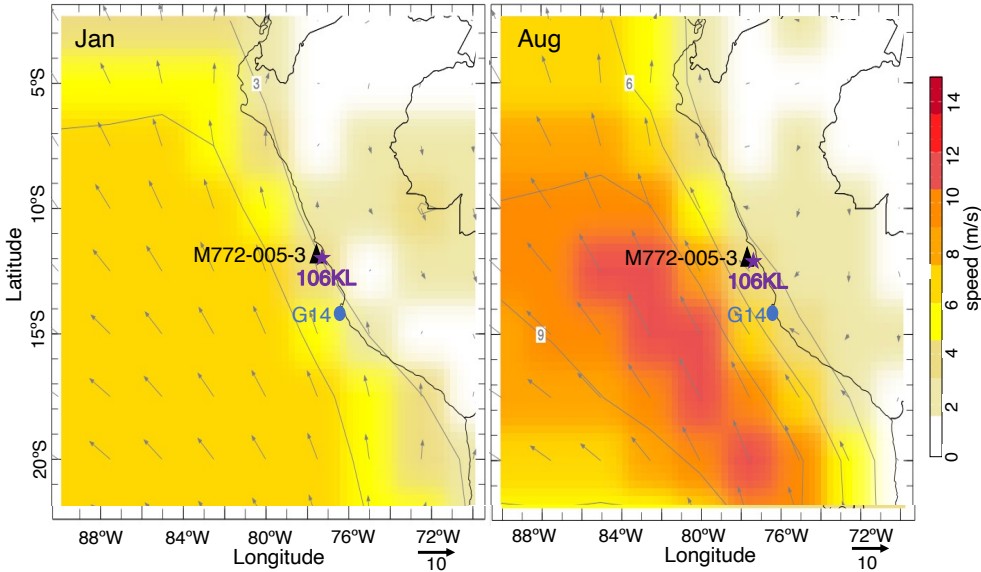

**Figure 1. (A) Location of the sampling of the sediment cores M77/2-005-3, 106Kl and G14 core. Average wind speed (m/s) at 1000 hPa in January and August (http://iridl.ldeo.columbia.edu).**


## 2.2 Grain-size distribution and end-member analysis

In order to reduce the effect of resuspension of particles that can produce artificial results, only laminated and banded sequences were subsampled for grain-size distribution analysis. Reworked sediments are widespread in the marine sediment records off Peru and can be distinguished from well-preserved sediments using X-ray images (Salvatteci et al., 2014b). For both surface

and core sediment samples, in order to isolate the terrigenous fraction, we followed the procedure described in Briceño-Zuluaga et al. (2016). Organic matter, calcium carbonates and biogenic silica were successively removed with hydrogen peroxide ($H_2O_2$ 30% at 60ºC for 3 to 4 days), hydrochloric acid (HCl 10% for 12h) and sodium carbonate ($Na_2CO_3$, 1M at 90ºC for 3h), respectively. The grain size distribution was then measured using an automated image analysis system (model FPIA3000, Malvern Instruments) with a measurement range of 0.5-200 µm. Further details on the FPIA3000 are described in Flores-

Aqueveque et al. (2014) and Briceño-Zuluaga et al. (2016). Given that only particles smaller than 200 µm can be measured under the analytical conditions applied in this work, all samples were sieved with a 200 µm mesh before being analyzed. Particles larger than 200 µm were not recovered in any sample, thus we consider that our analysis covers the full range of grain size present in the surface and cores sediments.

AnalySize modelling algorithm (Paterson and Heslop, 2015), which is based on the unmixing performed in hyperspectral

image analysis, was used to unmix the grain-size distribution. The algorithm claims to establish a physical mixing model that transforms the measured grain-size distribution to a limited number of unimodal grain-size end members. The coefficient of





determination ($r^2$) which represents the proportion of the variance of each grain size class is calculated to estimate the minimum number of end members necessary. More specific details are available from Paterson and Heslop (2015).

### 2.3 XRF analysis

The piston core KL 106 was analyzed with an Avaatech XRF Core Scanner at the Marum Center (Bremen). Here XRF scanning was done for 19 elements at 2-mm intervals. Ti is an aluminum/silicate-related element and is associated with clay minerals transported from the continent to the ocean through river discharges (Jansen et al., 1998; Yarincik et al., 2000). For this reason, Ti is employed as a proxy for river discharge (e.g., Haug et al., 2001). In contrast, Zr is predominantly enriched in heavy mineral species, in particular zircon. The latter is broadly distributed in natural sediments and typically has relatively coarse

grain size (Pettijohn, 1941). Zr has been widely used in studies such proxy of mean depositional grain-size variations (e.g., Dypvik and Harris, 2001; Wu et al., 2020). Therefore, Zr-rich sand can be used as a potential proxy for aeolian input. As a consequence of the above, we used the logarithm of the ratio of Ti to Zr intensity (counts per second) as a potential proxy for fluvial versus aeolian input.

### 3. Results

**3.1 Grain size distribution of the surface sediments**

The grain-size distribution of all surface samples collected at the stations in Callao and Pisco are shown in Figure 2. In Callao and Pisco, during normal conditions, the abundance of fine particles (< 10 μm) is higher at the deepest and furthest off shore station than at coastal stations, while, coarse particles (60-120 μm) are more abundant close to the coast (Fig. 2). Coarser particles are deposited mainly on the inner continental shelf due to their weight. During the 2017 Coastal El Niño, a strong

increase in fine particles (< 10 μm) abundance was found only at station E2 (Fig. 2a) in Callao. On the other hand, the abundance of the coarse fraction (50-100 μm) in the sediments from the most distant stations (E12 and E11) increased in Pisco (Fig. 2d and 2e).





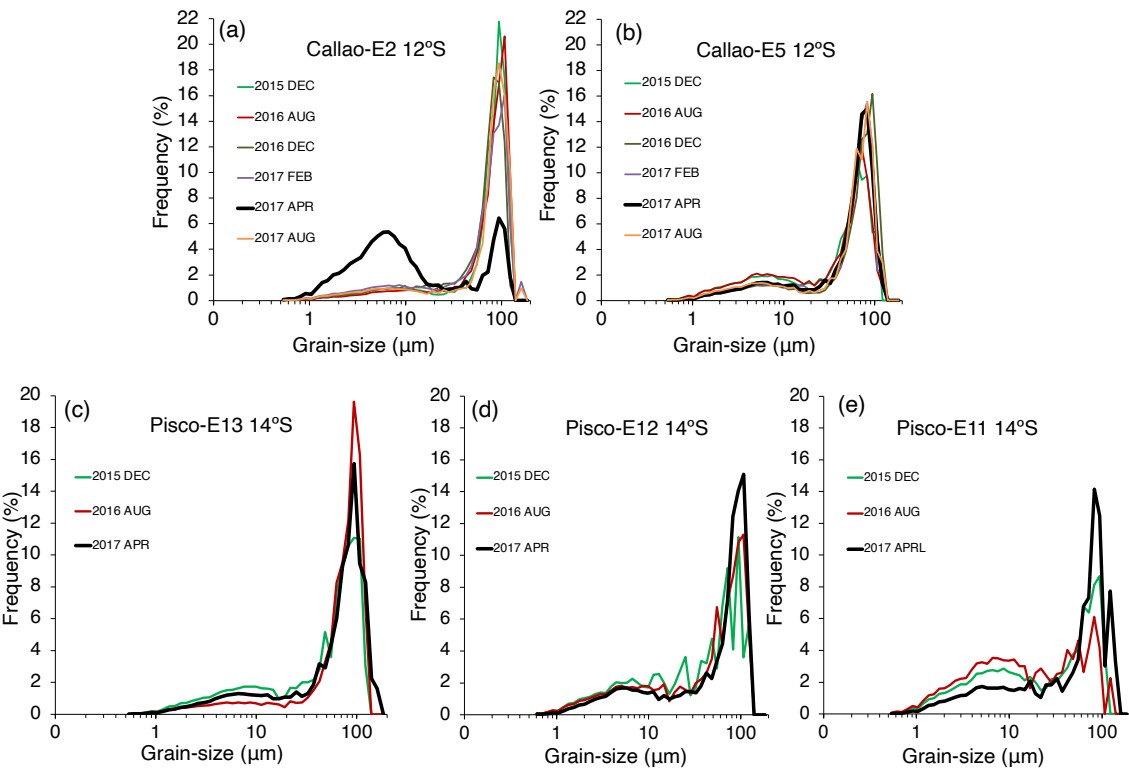

**Figure 2. Grain-size distribution of surface sediments in Callao (a, b) and Pisco stations (c, d, e).**

The mean grain-size distribution per climatic period analyzed is shown in figure 3a and 3b. During the Late HS1 (16-14.7 kyr BP), the abundance of fine particles (< 10 μm) is higher than during the Early HS1 (18-16 kyr BP) and B-A in Callao and Pisco (Fig. 3a and 3b). This increase is more marked in Callao (Fig. 3a). On the other hand, during the B-A (14.7-13 kyr BP), the sediments are characterized by high abundance of coarser particles (Fig 3a and 3b).

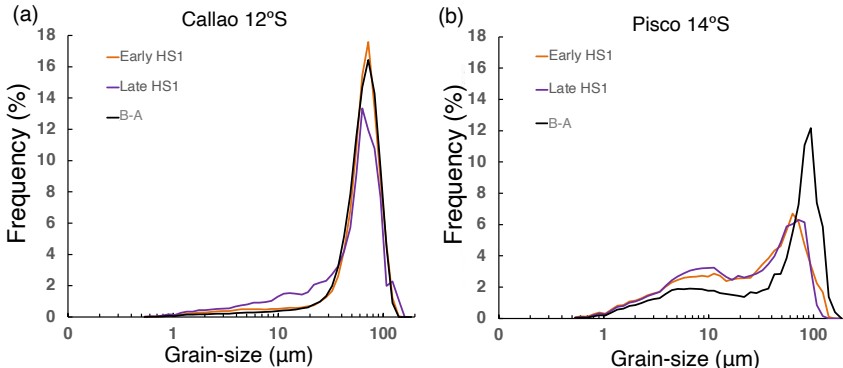

**Figure 3. Mean Grain-size distribution by climatic period in Callao (a) and Pisco (b).**



### 3.3 End-member analysis

Based on a multiple correlation coefficient, a model with 4 end members was chosen in Callao and Pisco, explaining 98% and 95% of the variance of the grain-size distribution data set respectively (Fig. 4a and 4b). The measured and modeled grain size distribution are highly correlated ($R^2$: 0.86-0.99) for each analyzed sample, attesting that the use of 4 end-members is appropriate for our interpretation. Although a model with 2 end-member model explains 95% of the variance of the data in Callao, the variability of each end-members contribution is different (Fig. 5) suggesting that each end member indicates different processes or sources (Fig. 5). Each of the 4 end members (EM) presents a unimodal distribution, its median being respectively at 11 µm (EM1), 58 µm (EM2), 77 µm (EM3) and 101 µm (EM4) in Callao (Fig. 4c). In Pisco, each EM is represented by a unimodal distribution centered at 10 µm (EM1), 52 µm (EM2), 75 µm (EM3) and 94 µm (EM4) (Fig. 4d).

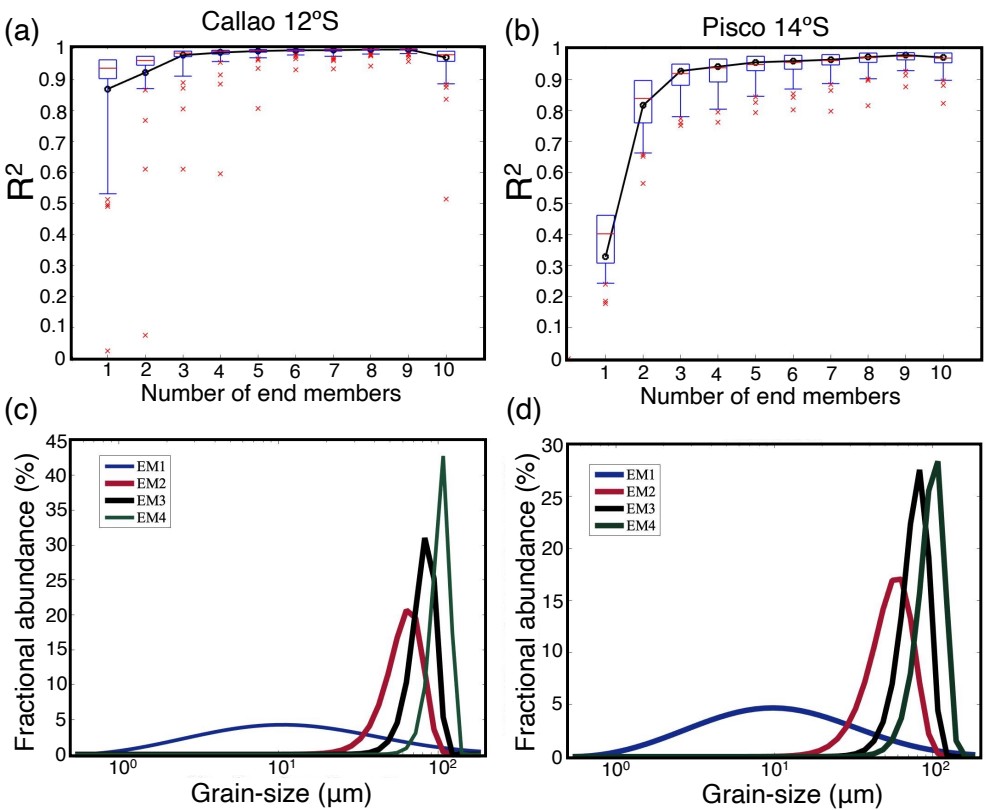

**Figure 4. Coefficient of determination (r2) as a function of the number of end members chosen to model the observed grain-size distribution in Callao (a) and Pisco (b). Grain size distribution of the 4 end-members in Callao (c) and Pisco (d).**



### 3.4 XRF analysis

225 In Callao, the Ti/Zr record from core 106KL indicate an increase of fluvial discharge during the HS1 with higher fluvial discharge between ~15.5-14.9 kyr BP and decrease of fluvial inputs afterwards, during the B-A (Fig. 6f).

## 4. Discussion

### 4.1 Assignment of end-members

The terrigenous materials deposited on the Peruvian margin are transported by rivers and by wind activity, however, there are 230 also other diffused sources, such as the material produced by coastal erosion transported offshore by marine currents. The terrigenous sediments in Callao and Pisco are multimodal which suggest that different processes are involved in the transport and deposition of these sediments. In both cores, we observed one end-member corresponding to a fine fraction (EM1) and three end-members to coarse fractions (EM2, EM3 and EM4). A previous study in Pisco used the variations of fine (10 μm) and coarse (50 μm and 100 μm) particles as proxies of river discharge and aeolian inputs respectively for the last few centuries 235 (Briceño-Zuluaga et al., 2016). However, the differences in the aeolian sources, shelf and slope morphology, currents intensity and hydrodynamics between Callao and Pisco, may modify the interpretation of the proxies described by Briceño-Zuluaga et al. (2016).

EM1 shows a mode at 11 μm and 10 μm in Callao and Pisco respectively that we interpret as a proxy for fluvial source, based on our interpretations and the literature (e.g., Stuut and Lamy, 2004; Stuut et al., 2007; Briceño-Zuluaga et al., 2016; Beuscher 240 et al., 2017). Previous studies in the Pisco area for the last few centuries suggest that abundance of fine particles (~ 10 μm) can be used as a proxy for fluvial inputs and precipitation patterns (Briceño-Zuluaga et al., 2016). The increase of fine particles (<10 μm) and EM1 contribution in surface marine sediments (Fig. 2a and Fig. 5a) associated with local high fluvial discharges in Callao during the 2017 Coastal El Niño (Guzman et al., 2020) corroborates the use of variations in the abundance of fine particles as an indicator of fluvial inputs. These results are consistent with the grain size of fine particles (~ 6-14 μm) in marine 245 cores associated with river inputs reported in different areas of the world (e.g., Stuut and Lamy, 2004; Stuut et al., 2007; Beuscher et al., 2017).

Previous studies in South Eastern Pacific used the abundance and fluxes of coarse particles (~ 36-100 μm) in marine sediments as a proxy for wind intensity linked to the expansion/contraction of the SPSH (e.g., Flores-Aqueveque et al., 2015; Briceño-Zuluaga et al., 2016). Based on HYSPLIT (Hybrid Single-Particle Lagrangian Integrated Trajectory) simulations, Briceño-250 Zuluaga et al. (2017) showed that coarse particles (50-90 μm) can directly reach the continental shelf in Pisco during Paracas storms (characterized by wind velocities surpassing 10-15 m/s).

The end members associated with the coarse fraction present similar modes in Callao and Pisco sediments (EM2, ~ 55 μm, EM3, ~ 75 μm and EM4, ~ 90-100 μm). Pisco is characterized by a large coastal desert and frequent dust storms. Also, since the shelf at Pisco is narrow, the distance from terrestrial sources that can be transported by winds did not vary significantly





during the deglaciation compared to modern conditions. Based on this, particles between 50-90 μm (EM2, EM3 and EM4) are interpreted as indicators of aeolian input, as proposed by Briceño-Zuluaga et al. (2016). An increment of the coarse particle abundance in Pisco's surface sediments was recorded in the stations most distant from the coast E12 and E11 during April 2017 (Fig. 2d and 2e), when alongshore wind stress was anomalously enhanced at the mature phase of the Coastal El Niño, particularly off central and southern Peru (Echevin et al., 2018). This observation supports the hypothesis that these particles

have an aeolian origin and the increase of their contribution suggests that during events with stronger alongshore winds, these particles can be transported to a greater distance than during normal conditions, modifying their amplitude in the sediments. It is important to note that the number of surface sediment samples in Pisco was small, however a change in the abundance of the coarse fraction was recorded during April 2017 (Fig. 2d and 2e). Finally, although EM2, EM3 and EM4 reflect an aeolian source, their contributions variations during the last deglaciation were different (Fig. 5b). This is possibly explained by changes

in wind intensity. Periods with stronger (weaker) winds result in an increase (decrease) in the amplitude of coarser particles. In Callao, the context is different, since there are no large desserts near the coast and dust storms are rare, it is unlikely for ~ 100 μm particles (EM4) to be transported directly to the sampling site. The presence of ~100 μm particles is possibly linked to bottom hydrodynamic processes. Indeed, during events of higher hydrodynamic energy, resuspension of fine particles and increase in the relative frequency of coarser particles are expected. EM4 (101 μm) does not show drastic changes during the

last deglaciation, suggesting that it does not influence the relative contribution of the other modes at millennial-scale (Fig. 5a). EM2 (58 μm) is the dominant mode in Callao ranging from 40 to 80% (Fig. 5a). Although these coarse sediments can be transported by winds (Briceño-Zuluaga et al., 2016, 2017), it is unlikely that this high percentage of coarse particles could be solely transported by the wind directly to the sampling site off Callao. It is more likely that ~ 58 μm particles derive from different sources (winds, coastal erosion) and are distributed on the continental shelf by the Peru-Chile Undercurrent

(Reinhardt et al., 2002).

During the Coastal Niño April 2017 event, we observed that at the E5 station (55 km offshore from the coast line), only the contribution of ~ 77 μm particles (EM3) increased (Fig. 5a), probably as a result of more intense alongshore winds. On the other hand, the relative stability of the EM2 (58 μm) and EM4 (101 μm) contributions show that these modes are not sensitive to wind intensity and probably respond to other processes (Fig. 5a). On the opposite, EM3 appears as the best proxy of a wind

source. In summary, in Callao, we propose that changes in the contribution of EM3 suggest changes in aeolian inputs and wind intensity, while changes in the contribution of EM2 and EM4 are associated with other processes associated with diffuse sources and hydrodynamic energy.



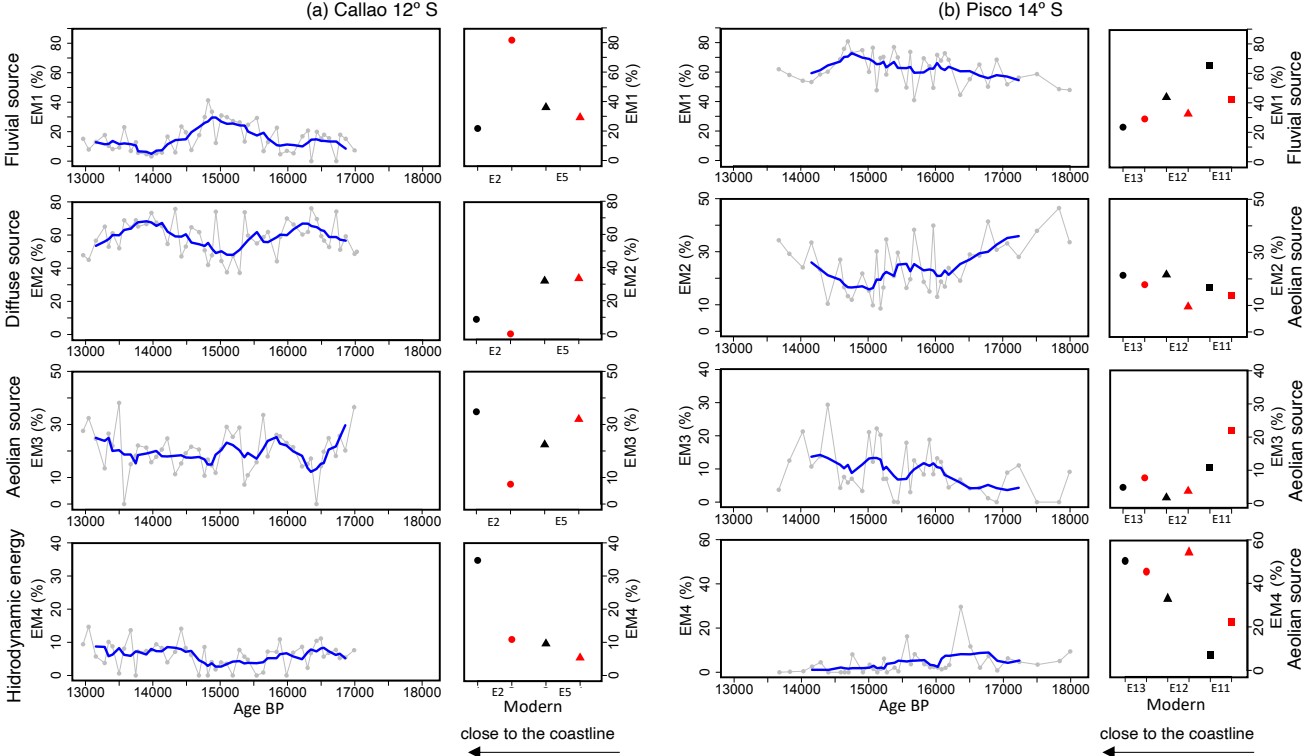

**Figure 5. Variation in contribution of the grain size end-members from marine cores and surface sediments in Callao (a) and Pisco (b). The Modern period is represented by the mean end-member contribution of surface sediments collected during normal conditions (black symbols) and Coastal El Niño April 2017 event (red symbols).**

## 4.2. Millennial variability of fluvial and aeolian inputs during the last deglaciation

The analysis of end-members allows a quantification of the main granulometric modes, however, as the sum of the contribution of the modes corresponds to 100%, it is difficult to consider the modes individually since the increase of one may reduce the others and influence their contribution variability. Thus, for a better visualization, a ratio between end-members indicative of a fluvial (EM1 in Callao and Pisco) and aeolian (EM3 in Callao and the sum of EM2, EM3 and EM4 in Pisco) source will be used as a proxy of the variations in the fluvial and aeolian inputs.

An increase in fluvial inputs based on the grain-size and end-member analysis was observed during the Late HS1 (~16-14.7 kyr BP) with maximum values between ~15.5-14.7 kyr BP in Callao (M77/2-005-3) and Pisco (G14) (Fig. 6g and 6h). Likewise, the Ti /Zr record in Callao (106KL) indicates an increase in fluvial input during HS1, with maximum discharges between ~15.5-14.9 kyr BP. These observations support the idea that Ti, linked to the fine fraction of marine sediments, is mainly transported by rivers to the Peruvian margin and can be used as a proxy of river discharge and precipitation (e.g., Mollier-Vogel et al., 2013; Salvatteci et al., 2014; Fleury et al., 2015).



The contrasting differences between our record of fluvial input (based on grain size and Ti/Zr) and the record of lithic content based on reflectance (Rein et al., 2005) in Callao can be explained by the difference in methodology and interpretation of the proxies. Rein et al. (2005) interpret the lithic content as a proxy for river discharges, however, as observed in our data (Fig. 5) and the literature (Briceño-Zuluaga et al., 2016), the terrigenous material can be transported to the central-southern Peruvian margin by different sources (e.g., fluvial and aeolian) and the variability of fluvial and aeolian transport follow different

patterns and therefore respond to different forcing.

Since heavy precipitation associated with the El Niño events in Callao and Pisco coastal regions are occasional, a larger average fluvial discharge in Callao and Pisco is likely related to precipitation fluctuations at higher elevations in the watersheds, in the Andes. Previous studies suggest a correlation between North Atlantic cooling and massive meltwater discharges with increased precipitation in the Central Andes (Baker et al., 2001a, 2001b; Blard et al., 2011, Martin et al., 2018; González-Pinilla et al.,

2021). During the last deglaciation, cooling in the North Atlantic and higher meltwater discharges generated a weakening of the AMOC (MacManus et al., 2004; Mulitza et al., 2017; Ng et al., 2018). The latter generated an interhemispheric temperature contrast and an impact on precipitation in the central Andes associated with a southward shift of the ITCZ and an intensification of the South American Monsoon in different regions: Central Andes (Baker et al., 2001a, 2001b, Blard et al., 2011; Gonzalez-Pinilla et al. 2021), southeast (Cruz et al. 2005; Stríkis et al., 2015) and southwest Brazil (Novello et al., 2017) and in western

Amazonia (Sublette Mosblech et al., 2012; Cheng et al., 2013). Indeed, the higher river discharges we evidenced in Callao and Pisco during Late HS1 (~16-14.7 kyr BP) occurred simultaneously with the well-dated highstand of the giant paleolake Tauca (~16.6-14.5 kyr BP) (Martin et al., 2018). Therefore, the increase and decrease of river discharges in central Peru during HS1 and B-A, respectively, would be explained by changes in precipitation in the Andes in response to changes in the intensity of the AMOC and meltwater pulses in the North Atlantic.

During the last decades, the seasonal ITCZ have shifted poleward in South Pacific and generally narrowed and strengthened (Zhou et al., 2020). A recent study suggests a narrowing and southward shift of ITCZ in Eastern Pacific in response to the SSP3-7.0 scenario by 2100 (Mamalakis et al., 2021). Although there are uncertainties about the effects of current global warming on AMOC intensity, there is evidence of AMOC slowing over the last century (Rahmstorf et al., 2015; Caesar et al., 2018) and in climate model simulations of future climate change, AMOC is projected to decline generating a southward

displacement of the ITCZ (Bellomo et al., 2021). In the context of global warming, a large-scale precipitation and fluvial discharge increases in Peru related to AMOC decline and southward displacement of the ITCZ should be considered.

Concerning the variations of the aeolian inputs and surface wind intensity in the Peruvian margin, changes in the intensity of Walker Circulation and meridional displacements of the ITCZ-SPSH system have been proposed as mechanisms to regulate

surface alongshore winds and upwelling dynamics in the Humboldt Current System at multiple timescales (e.g., Gutiérrez et al., 2009; Briceño-Zuluaga et al., 2016; Salvatteci et al., 2014, 2019). On centennial timescales, a northern displacement of the SPSH-ITCZ, in response to stronger Walker circulation produce an increase of alongshore winds and upwelling in Central-South Peruvian margin (Salvatteci et al., 2014; Briceño-Zuluaga et al., 2016). SST proxies indicate an indicate an increase of





the zonal SST gradient increase in the equatorial Pacific indicates a more intense Walker circulation in B-A than during HS1

(Fig. 7a) (Koutavas and Joanides, 2012). Moreover, a northern displacement of the ITCZ was also recorded during B-A. (Peterson et al., 2000; Deplazes et al., 2013). These conditions should have provoked the increase of alongshore winds and aeolian supply in central Peru during B-A. Indeed, in Callao and Pisco, the aeolian inputs was the main transport from 14.7 to 13 kyr BP suggesting, at regional-scale, an increase of alongshore wind and upwelling in Central-South Peru and an expansion of the SPSH (Fig. 7d and 7e). Our record of surface wind intensity variations and an alkenone-derived SST reconstruction

based on alkenones (Salvatteci et al., 2019) in the same core collected in Callao show similar trends (Fig. 7c). During HS1, a short cooling between 16 and 15.5 kyr BP, coincided with stronger alongshore winds (Fig. 7c and d). In addition, during the B-A, a cooling at 14.7-13 kyr BP, stronger aeolian transport associated with more intense alongshore winds occurred (Fig. 7c and 7d). These observations suggest that local processes such as upwelling variations in response to changes in alongshore wind intensity may control SST variations during the last deglaciation, in addition to other process like the advection of the

Southern Ocean and Antarctic climate signals by the Humboldt Current.



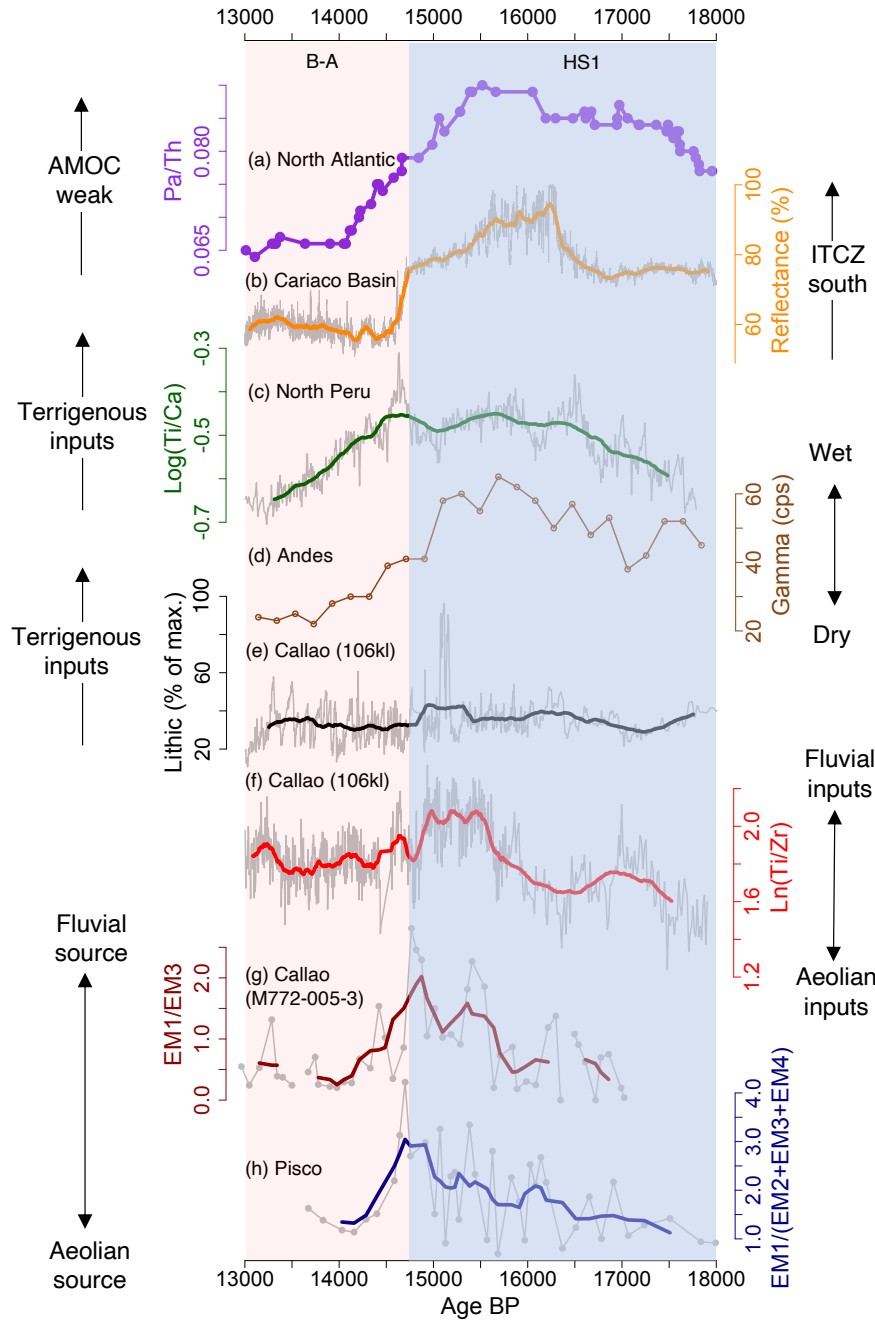

**Figure 6. (a) Composite 231Pa/230Th record that reflect past changes in AMOC (Ng et al., 2018). (b) Reflectance (%) from Cariaco Basin as a proxy for the latitudinal displacement of the ITCZ (Deplazes et al., 2013). (c) Log (Ti/Ca) for core M77/2-059 from northern Peru (4° S) (Mollier-Vogel et al., 2003). (d) Natural γ-radiation as a proxy for effective moisture in Tropical Andes (Baker et al., 2021b). (e) Relative concentration of lithics for core 106Kl from Callao (Rein et al, 2005). (f) Ln (Ti/Zr) as a proxy for fluvial vs aeolian inputs in core 106KL from Callao (this study). (g) EM1/EM2 ratio as a proxy for fluvial and aeolian source in Callao (this study). (h) EM1/(EM2+EM3+EM4) ratio as a proxy for fluvial and aeolian source in Callao (this study).**


**Climate of the Past** Discussions — Open Access — EGU

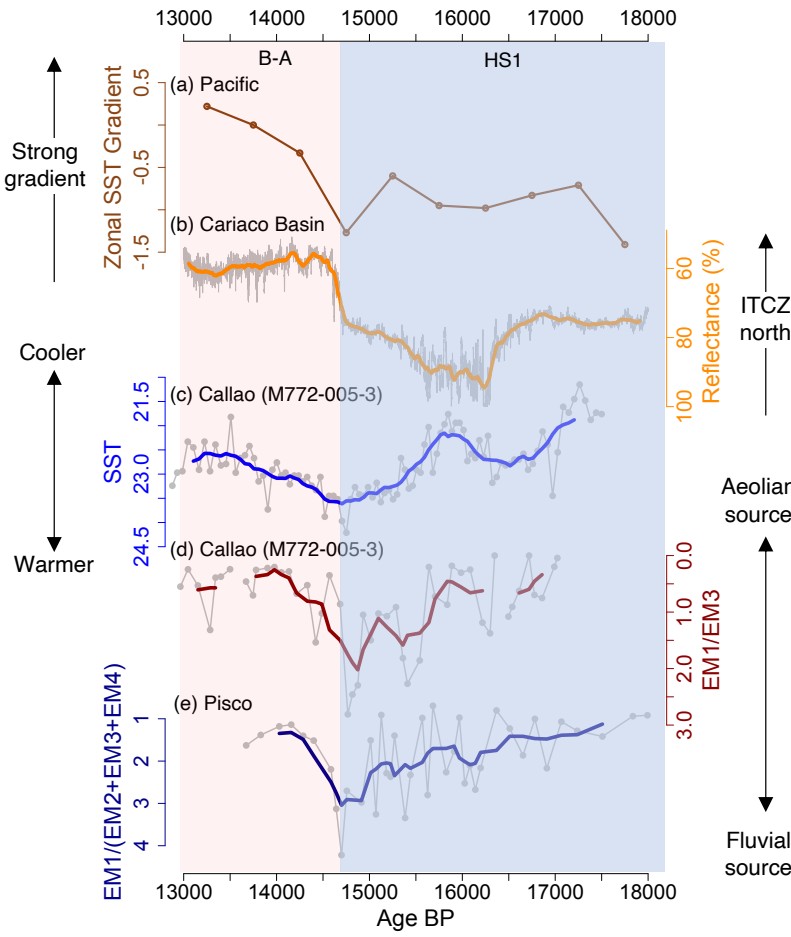


**Figure 7. (a) Zonal SST gradient anomaly during the last deglaciation calculated as the difference between western and eastern Pacific averages (Koutavas and Joanides, 2012). (b) Reflectance (%) from Cariaco Basin as a proxy for the latitudinal displacement of the ITCZ (Deplazes et al., 2013). (c) Alkenone-derived near surface temperature from M77/2-005-3 core, Callao (Salvatteci et al., 2019). (d) EM1/EM2 ratio (Reversal scale) as a proxy for fluvial and aeolian source in Callao (this study). (e) EM1/(EM2+EM3+EM4) ratio (Reversal scale) as a proxy for fluvial and aeolian source in Pisco (this study).**


## 5. Conclusion

The variability of the grain size distribution of marine sediments from the central-southern Peruvian margin (12ºS and 14ºS) reveals millennial-scale changes in the transport and sedimentation processes of the terrigenous material during the last deglaciation (18-13 kyr BP). We identified four granulometric end-members for both Callao and Pisco sediments, each of

them reflecting different processes and sources and whose interpretation must consider regional contexts. In the case of the Pisco core, located within the range of the aeolian inputs, as it has been shown earlier, the modes (EM2 to EM4) correspond to aeolian origin. In the case of Callao core, located further from the coast and where sources of eolian particles are scarce, the



EM2 and EM4 modes are interpreted as reflecting local hydrodynamics, while EM3 represents the eolian supply. Our results support a tight relationship between high latitude forcing and precipitation in the western flank of the Andes during the last

deglaciation. During late HS1 (16-15 kyr BP), enhanced fluvial inputs in Callao and Pisco occurred associated with higher precipitation in Central Andes in response to the slowdown of AMOC and meltwater discharge in North Atlantic. Finally, the increase in aeolian input during the B-A, could be a result of stronger alongshore winds linked to a northern displacement of the ITCZ-SPSH system in response to a strong gradient of the Walker circulation. There is still uncertainty about the effects of current Climate Change, there is evidence of a slowing of the AMOC over the past century and in future climate model

simulations. In the latter, the decline in AMOC is accompanied by a southward shift in the ITCZ. Thus, we can probably expect an increase in precipitation and river flow in the future in Peru.

Data availability

The data associated with this manuscript will be submitted in the PANGAEA database upon publication of this paper.


Author contributions

MY, BT and DG designed the study. MY and SC carried out the grain-size analysis. DG and GS conducted the XRF analysis of core KL 106. MY wrote the manuscript with the help of BT and SC. All authors discussed and commented on the paper.

Competing interests

The authors declare that they have no conflict of interests.

Acknowledgements

This publication was made possible through support provided by the IRD-DPF and the MAGNET Program of CONCYTEC

N°007-2017. This work was supported by the International Joint Laboratory "PALEOTRACES" (IRD, France; UPMC, France; UFF, Brazil; UA, Chile; UPCH, Peru) and ANR-15-JCLI-0003-03 BELMONT FORUM PACMEDY. Grain-size analyses were performed on the ALYSES facility (IRD-Sorbonne University, Bondy, France), which is supported by grants from Région Ile-de-France. This work is a contribution of the Collaborative Research Project 754 "Climate-Biogeochemistry interactions in the Tropical Ocean" (www.sfb754.de), which is supported by the Deutsche Forschungsgemeinschaft (DFG).

We would like to thank the crew and scientists aboard R/V Meteor cruises M77/2 in 2008. We deeply thank Bo Thamdrup, chief scientist of the Galathea-3 expedition (Leg 14), and Bente Lomstein, who conducted the core sampling onboard the RV Vaedderen.



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
