# Peer review of "Millennial variability of terrigenous transport to the central-southern Peruvian margin during the last deglaciation (18-13 kyr BP)"

_Climate of the Past, 2021_

## Author Comment (AC1)

[Figure]

Fig1. Callao and Pisco rivers flows during El Niño 2017.

[Figure]

Fig 2. Wind surface anomalies during April 2017 from ASCAT

---

## Author Response (AR1)

Dear Editor,

We follow all the recommendations of the reviewers excepted a little change suggested, if possible, on fig. 6 and 7 that we could not perform by lack of modern data.
In addition, we performed new ICPMS analyses to obtain Ti and Zr concentrations in surface sediments. For this reason, we have associated a new author: Federico Velazco who collected these sediments, prepared them and helped to interpret them. These analyses clearly show that Ti is associated with fine sediments of fluvial inputs during the 2017 Coastal El Nino. This greatly enriches the discussion and solidifies our interpretation.
We are very pleased with this new version and thank you and the two reviewers for being very helpful in improving the manuscript.

Our comments are written in read. References to new lines number of the corrections described below correspond to line number in the final version of the manuscript.

Reviewer 1

The article is generally clearly written, but the quality of the english is variable throughout the manuscript. The text could be somewhat condensed to make the reading easier.

We thank the reviewer for the constructive comments. As suggested by the reviewer, an attempt was made to condense the text and we contract a proofreading service to improve the English. These proofreading corrections have not been included in the word document which shows the other corrections described below for clarity.

Detail comments:

L97: at what depth are the anoxic conditions?

The information was added to the new manuscript line 105.

L110: precise "austral" winter and "austral" summer

Done line 123 and elsewhere in the document.

L113: "large quantities" seems in contradiction with L115 "scarce flows of coastal rivers".

That true, the text was modified according to the reviewer's suggestion lines 121- 1124

L119: "Aeolian clays and silts are transported offshore by trade winds" implies that the fine fraction is not only related to fluvial transport but also partly to aeolian transport, which is at odds with the conclusions. A clarification is required.

We now explain that "A large fraction of the particles smaller than 10 µm (desert aerosols) are transported beyond the continental shelf (Saukel et al., 2011). Therefore, the fine fraction of aeolian origin in the continental shelf sediments is therefore negligible, and the fine fraction of the continental shelf is largely dominated by fluvial inputs." Line 128-130.

L128: please indicate the number of samples and the range of depth in the main text.

Done line 137-139.

L130: 2017 coastal El Niño anomalies did not extend south of Lima. "El Niño" conditions may thus not be justified for Pisco.

More details on the El Niño 2017 was added to the new manuscript. L. 86-93

L140: which calibration dataset does CALIB8.1 use? Base on the figure, I understand that the depth-age model is a sequence of linear models. This should be explained in the method section.

The 14C ages and calibrated ages are listed in Supplementary Table 2. CALIB 8.1 used the Marine 20 dataset now mentioned in the text line 150-151.

L149: "laminated packaged"? do you mean "lamination"?

We change for laminated sediments.

L154: the relevant information is the 14C calibration dataset, not the software.

We used the 14C ages published in Rein et al (2005) and calibration was by CALIB 8.1 with the Marine 20 dataset as now mentioned in the text.

L174-178: this paragraph needs to be clarified, in part by improving the english.

The text was modified according to the reviewer's suggestion lines 185-190.

L181-182: What about wind blown terrestrial material? Doesn't it contain Ti as well? In Haug et al 2001, Ti was used indeed as a proxy for river discharge but aeolian transport was not an issue there.

We now mention that "Using the Ti/Zr ratio, we compared Ti, which is present in all sizes of sediment, but especially in clays, to Zr, which would only be present in the silt and sand fractions in the form of zircon minerals. Assuming that fine particles such as clays are mainly transported by rivers, while coarse particles (coarse silts and sands) come mainly from an aeolian origin, the Ti/Zr ratio can be used as a potential proxy for fluvial versys aeolian inputs." Lines 205-209.

L185: "Zr has been widely used as a proxy for mean depositional grain-size"

Same modification as above.

Figure 2: add the sample depth to each graph.

The sample depth has been added to each graph as suggested by the reviewer.

L238: "we interpret...based on our interpretation". So, what is the interpretation based on?

This paragraph was rewritten. L 274-281.

L238-246: This paragraph should be shorter. The absence of a fine fraction peak in april 2017 in station E5 should be discussed.

This paragraph was rewritten. L 274-281 where we also explain that "The non-increase of fine particles in E5 Callao during Coastal El Niño is probably due to the greater distance between the fluvial source and the sampling site." (Line 279-278).

L258: "when alongshore wind stress was anomalously enhanced at the mature phase of the Coastal El Niño". This contradicts earlier statement (introduction) about the increase of rainfall during El Niño. More details about the 2017 El Niño event are needed.

Increase in surface winds and precipitation occurred simultaneously during El Niño 2017. More details on the El Niño 2017 were given in the introduction.

L262-263: I don't understand clearly what the authors mean.

We specify that: "because of t the small number of samples, this variation was not statistically significant." L. 296-297

L266: "desert" instead of "dessert"

OK, the responsible author will be deprived of dessert. L. 301

L276-282: keeping EM3 as a wind proxy and excluding EM2 and EM4 only on the base of the slight increase of EM3 in one station in April 2017, seems weak. EM4 increased in Pisco in April 2017. The argument needs to be strengthened. The whole section in general is somewhat long and confusing.

We shortened this section and rewrote the paragraph L. 310-313.

L292: I did not understand from the previous section that EM2+EM3+EM4 would be considered as wind proxy in Pisco. This needs to be more clearly built and stated in section 4.1.

That was also the objective of this new paragraph cited above.

L296-297: a similar result would be obtained if Ti was both in fluvial and aeolian material, and Zr only in aeolian.

We now present ICPMS analysis showing the Increase of Ti concentration in April 2017 at E2 station showing the high Ti concentration in fluvial sediments (Fig. 5 and lines 329-333).

L320: "seasonal": which season? "poleward": North or South?

"During the past few decades, the ITCZ in East Pacific have shifted southward…" L 354

L333-334: "SST proxies..." please correct and clarify the sentence.

Part of the sentence has been removed (L. 367)

Figure 6 and 7: when possible, indicating a modern value for reference would be useful.

We agreed with the reviewer but the problem is that we only have modern values for the Callao site, and even there the 6 samples analyzed in the surface sediments of this site show a great temporal variability. So we could not follow this suggestion.

Supplementary Figure 1: legend for triangles is missing

The legend for triangles was added in the new manuscript.

supplementary figure 2: please add legend for the symbols. What are the stars? Dates from Salvatecci? Are they included in the calculation of the new age model? Indicate sedimentation hiatus.

As the reviewer says, the stars are the ages of Salvatteci et al. (2019). These ages were taken into account in the new age model and are listed in the table in the supplementary information. In the new manuscript the corresponding legends was added.

Reviewer 2

Minor changes could improve the quality of the text, and besides the suggestions made by RC1, which I totally agree with, below there are some suggestions:

We thank the reviewer for the constructive comments.

Authors could present a distribution map for the surface samples stations.

A map indicating the location of the surface samples was presented in supplementary information and new maps were prepared in the new manuscript.

In methods, they mention the de 14C analyses were performed in bulk samples, but they did not mention if the organic matter or carbonate were dated, since they use the delta R to calibrate, I assume that they have dated the carbonates of the sample. But it should be mentioned, and why they chose to date the bulk sample and not just specific forams e.g.

It is now mentioned that the 14C analyses were performed on organic matter. L. 147

In topic 3.4, the authors could better explore the XRF result, which is very important to support the end members. Ideally, the results section should not present interpretation, and as it stands, this topic only presents the interpretation of changes in fluvial discharge.

Topic 3.4 has been rewritten now with the new support of ICPMS Analysis of surface sediments.

Line 23: change the font after 77 to symbol, from "u" to "m".

Done at line 23.

Line 41: Overturning is misspelled.

Done Line 42.

Line 85: "the" is missing before "other".

This sentence is no more in the manuscript.

Line 96: I suggest using the past tense.

Done line 103.

Line 138: ad "s" after "period" and remove "d" from calibrated.

OK, Line 149.

Line 185: review the use of the word "such"

Corrected line 204.

Line 193: offshore is misspelled

Corrected line 213.

Line 240: "the" is missing before "abundance"

This paragraph has been rewritten.

Line 333-334: remove the sentence: "SST proxies indicate an indicate an increase of the zonal"

The erroneous part of the sentence has been removed L.367.

Line 344: change "process" to "processes"

Done line 377.

Line 353: it is Pisco instead of Callao.

Corrected line 387.

Line 371: "the" missing before "North"

OK Line 5406.